# Paternal grandfather's access to food predicts all-cause and cancer mortality in grandsons

Denny Vågerö [1], Pia R. Pinger [2,3], Vanda Aronsson [1] & Gerard J. van den Berg [4,5]

Studies of animals and plants suggest that nutritional conditions in one generation may affect phenotypic characteristics in subsequent generations. A small number of human studies claim to show that pre-pubertal nutritional experience trigger a sex-specific transgenerational response along the male line. A single historical dataset, the Överkalix cohorts in northern Sweden, is often quoted as evidence. To test this hypothesis on an almost 40 times larger dataset we collect harvest data during the pre-pubertal period of grandparents (G0, n = 9,039) to examine its potential association with mortality in children (G1, n = 7,280) and grandchildren (G2, n = 11,561) in the Uppsala Multigeneration Study. We find support for the main Överkalix finding: paternal grandfather's food access in pre-puberty predicts his male, but not female, grandchildren's all-cause mortality. In our study, cancer mortality contributes strongly to this pattern. We are unable to reproduce previous results for diabetes and cardiovascular mortality.

---

[1] CHESS, Centre for Health Equity Studies, Department of Public Health Sciences, SE-106 91 Stockholm University, Stockholm, Sweden. [2] Department of Economics, University of Bonn, Adenauerallee 24-42, 53113 Bonn, Germany. [3] briq, Institute on Behavior & Inequality, Bonn, Germany. [4] Department of Economics, Priory Rd Complex, University of Bristol, Bristol BS8 1TU, United Kingdom. [5] IFAU, Institute for Evaluation of Labor Market and Education Policy, Uppsala, Sweden. Correspondence and requests for materials should be addressed to D.V. (email: denny.vagero@su.se)

Regulation of gene expression in response to environmental cues establishes a link between nature and nurture. Such a response may be carried over to the next generation as a so-called transgenerational response (TGR), and affect offspring phenotype. Epigenetically based TGR in mammals and plants has been shown[1-4]. Epigenetic changes transmitted through the germline in the Agouti mouse impacted five successive generations[5]. Evidence of TGR in humans is much more limited[6-8]. One of the problems in human studies is to separate genetic, epigenetic and cultural influences in transgenerational processes. This is particularly the case if they operate simultaneously and interact, which is very possible.

Recently, a series of much-cited papers based on the Överkalix cohorts in northern Sweden[9-12] has provided indirect evidence for a TGR. The authors reported that food shortage induced by harvest failures, or food abundance after a rich harvest, affected mortality outcomes in two subsequent generations. An intriguing aspect of these studies is the conjecture that an epigenetic pathway, carrying information across generations, may open up just before puberty, during the so-called slow growth period (SGP) [10,12]. Pre-puberty may be one of several "windows" for germline reprogramming[13,14] in response to nutritional signals. A number of mechanisms for transmission across generations have been suggested, usually involving DNA methylation, chromatin formation or small noncoding RNAs. After fertilization, in the preimplantation embryo, epigenetic modifications acquired early in life are then usually erased, but not fully[15]. Imprinting on specific loci may resist the post fertilization wave of reprogramming, eventually causing changes in offspring phenotype that are not driven by changes of the DNA sequence. Yehuda et al.[16] found that offspring of Holocaust survivors showed specific DNA methylation changes. Gapp et al.[17] demonstrated that traumatic stress in the early life of mice altered sperm microRNA expression, with behavioral and metabolic consequences in offspring. Rodgers et al.[18] found that paternal stress in mice altered sperm microRNA content, although this was not restricted to peripubertal stress. A transgenerational response, triggered during childhood, may also relate to an individual's genetic background, to changes in DNA sequence induced by environmental factors or due to chance events, such as de novo mutations in the germ line with the potential to change phenotypical characteristics in subsequent generations.

The findings from the Överkalix cohorts imply that grandparental access to food during their slow growth period can modify diabetes and all-cause mortality in grandchildren. Cardiovascular mortality on the other hand was associated with parental, but not grandparental, nutritional experience. The authors interpret their results[11] as "proof-of-principle that a sex-specific male-line transgenerational effect exists in humans", which they consider likely to be epigenetic rather than genetic, cultural or social. A summary of the Överkalix findings is available in[19]. Their findings have been discussed in renowned peer-reviewed journals[20-24] and are cited over 2,000 times (October 2018).

To establish evidence for a transgenerational response to early human nutritional experience, carried via the germline rather than based on direct in-utero exposure or maintained through continuity in social circumstances, we need to observe at least three generations and exogenous variation in the early nutritional conditions of the first generation. In addition, we need intergenerational social data. The Överkalix cohorts allow that kind of analysis, but suffer from other potential problems. Firstly, food access for six ancestors was examined in three small birth cohorts, where family ties were interwoven and rather complex. This has given rise to the concern that a multitude of comparisons have produced random or biased findings[25,26]. Secondly, since crop failures and very abundant harvests were rather infrequent, exposure to a nutritional shock during pre-puberty implies that individuals were born in particular years, leaving room for confounding due to birth cohort effects.

The current paper tests the hypothesis proposed in the Överkalix studies, that prepubertal nutritional conditions in one generation affect health outcomes in subsequent generations in a sex-specific manner. We, firstly, replicate all the Överkalix epidemiological analyses and, secondly, go beyond previously reported outcomes by also including cancer mortality in the grandchild generation. In this we are inspired by the work of Frankel et al.[27] who had demonstrated that abundant food in childhood was linked to elevated cancer mortality later in life. We use three linked generations (G0, G1, G2) of the Uppsala Birth Cohort Multigeneration Study. We follow the Överkalix approach in using the exogenous variation in harvests, based on crop statistics by region every year from 1874–1910 (details below). Easy or difficult access to food in a certain year and region is measured from this statistic. The validity of this method is discussed in the Methods section.

Our results support the hypothesis that a transgenerational response to abundant food in pre-puberty exists, along the male but not along the female line.

## Results

**Statistical power.** The most important results are given as Tables 1–3. Detailed results are given as Supplementary Tables 1 to 6. We estimated the effects of food access in generation 0 (G0) on mortality in their grandchildren (G2) and in their children (G1). All results refer to G0 food access during the slow growth period, defined as ages 9–12 for boys and 8–10 for girls[9].

The statistical power to find an association of at least the same strength as that reported from the Överkalix studies for each of their ten significant ($p < 0.05$) or near-significant ($p < 0.10$) results is shown in Supplementary Table 7. In analyses of G2 mortality this power varied from 72 to 99%. In analyzing G1 mortality, it was over 99%.

**Table 1 All-cause mortality 1961–2015 in G2 men by paternal grandparents' harvest exposures in SGP: hazard ratios with 95% confidence limits (in brackets) based on Cox regression**

| Access to food | All-cause mortality | | | |
| | Model 1 | | Model 2 | |
| --- | --- | --- | --- | --- |
| *Paternal grandmother* | | | | |
| Good | 0.88 | [0.46, 1.66] | 0.93 | [0.49, 1.76] |
| Intermediate | 1.00 | ref. | 1.00 | ref. |
| Poor | 0.68 | [0.36, 1.28] | 0.70 | [0.35, 1.37] |
| *Paternal grandfather* | | | | |
| Good | 1.50[a] | [0.99, 2.26] | **1.55**[b] | **[1.02, 2.35]** |
| Intermediate | 1.00 | ref. | 1.00 | ref. |
| Poor | 0.92 | [0.52, 1.62] | 0.93 | [0.51, 1.68] |
| *Observations* | 3224 | | 3224 | |
| *Number of deaths* | 339 | | 339 | |

Statistically significant estimates (95% CI) in bold type
Model 1: Adjusted for G2 birth year, sibship size and sibling order, father's harvest exposure in SGP, social class, income and education, and any parental death before age 18
Model 2: + linear trends for grandparents birth years, with confidence limits based on sibling cluster robust standard errors
[a]Interaction by gender: $p = 0.065$
[b]Interaction by gender: $p = 0.053$

**Table 2 Diabetes mortality 1961–2015 in G2 men and women by maternal grandparents' harvest exposures in SGP: hazard ratios with 95% confidence limits (in brackets) based on Cox regression**

| | Diabetes mortality | | | |
|---|---|---|---|---|
| Access to food | Model 1 | | Model 2 | |
| *Maternal grandmother* | | | | |
| Good | 2.25 | [0.86, 5.88] | **3.38** | **[1.18, 9.65]** |
| Intermediate | 1.00 | ref. | 1.00 | ref. |
| Poor | 0.96 | [0.22, 4.14] | 1.12 | [0.31, 4.07] |
| *Maternal grandfather* | | | | |
| Good | 0.46 | [0.06, 3.42] | 0.55 | [0.07, 4.61] |
| Intermediate | 1.00 | ref. | 1.00 | ref. |
| Poor | 0.52 | [0.07, 3.79] | 0.59 | [0.08, 4.26] |
| *Observations* | 5891 | | 5891 | |
| *Number of deaths* | 41 | | 41 | |

Statistically significant estimates (95% CI) in bold type
Model 1: Adjusted for G2 gender, birth year, sibship size and sibling order, mother's harvest exposure in SGP, social class, income and education, and any parental death before age 18
Model 2: + linear trends for grandparents birth years, with confidence limits based on sibling cluster robust standard errors

**Table 3 Cancer mortality 1961–2015 in G2 men by paternal grandparents' harvest exposures in SGP: hazard ratios with 95% confidence limits (in brackets) based on Cox regression**

| | All cancers | | | |
|---|---|---|---|---|
| Access to food | Model 1 | | Model 2 | |
| *Paternal grandmother* | | | | |
| Good | 1.09 | [0.39, 3.00] | 1.20 | [0.40, 3.62] |
| Intermediate | 1.00 | ref. | 1.00 | ref. |
| Poor | 1.37 | [0.59, 3.18] | 1.45 | [0.63, 3.34] |
| *Paternal grandfather* | | | | |
| Good | **3.35**[a] | **[1.95, 5.76]** | **3.44**[b] | **[1.87, 6.34]** |
| Intermediate | 1.00 | ref. | 1.00 | ref. |
| Poor | 0.65 | [0.20, 2.06] | 0.63 | [0.20, 1.99] |
| *Observations* | 3224 | | 3224 | |
| *Number of deaths* | 117 | | 117 | |
| ***Cancers not related to smoking*** | | | | |
| *Paternal grandmother* | | | | |
| Good | 0.00 | [0.00] | **0.00** | [0.00,] |
| Intermediate | 1.00 | ref. | 1.00 | ref. |
| Poor | 1.01 | [0.31, 3.28] | 1.13 | [0.35, 3.65] |
| *Paternal grandfather* | | | | |
| Good | **3.51**[c] | **[1.77, 6.97]** | **4.39**[d] | **[2.02, 9.53]** |
| Intermediate | 1.00 | ref. | 1.00 | ref. |
| Poor | 0.78 | [0.19, 3.26] | 0.86 | [0.21, 3.52] |
| *Observations* | 3224 | | 3224 | |
| *Number of deaths* | 70 | | 70 | |

Statistically significant estimates (95% CI) in bold type
Model 1: Adjusted for G2 birth year, sibship size and sibling order, father's harvest exposure in SGP, social class, income and education, and any parental death before age 18
Model 2: + linear trends for grandparents birth years, with confidence limits based on sibling cluster robust standard errors
[a]Interaction by gender: $p = 0.006$
[b]Interaction by gender: $p = 0.005$
[c]Interaction by gender: $p = 0.013$
[d]Interaction by gender: $p = 0.009$

**All-cause mortality**. If a paternal grandfather (G0) had unusually good access to food his grandsons (G2) appeared to have an elevated all-cause mortality (Table 1, model 1: HR = 1.50; 95% CI 0.99–2.26). This became clear when we adjusted for G0 birth year (model 2: HR = 1.55; CI: 1.02–2.35). In contrast, there was no excess mortality among his granddaughters (model 2: HR = 0.74; CI: 0.38–1.46; interaction by gender $p = 0.053$). We found (Supplementary Table 1) no link between either maternal grandparents' or paternal grandmothers' food access during their SGP and their grandchildren's all cause-mortality.

We tested sensitivity to the urban/rural factor by excluding all G0 who during their SGP were living in one of the then ten biggest cities in Sweden (seven of them are seaports). City inhabitants so defined constituted 17% of G0. Excluding them gave a somewhat higher estimate for grandsons with a paternal grandfather who experienced good harvest (model 2: HR = 1.68; CI: 1.02–2.78; interaction by gender $p = 0.056$).

Thus, our results suggest an elevated mortality among grandsons of paternal grandfathers with good access to food.

**Cardiovascular and diabetes mortality**. Food access in G0 was not significantly associated with cardiovascular disease mortality in grandchildren (Supplementary Table 2). We performed additional analyses combining hospitalizations and deaths with a CVD diagnosis as main or contributory cause, with the same result (Supplementary Table 5).

Table 2 suggests that a maternal grandmother who enjoyed good access to food during her SGP confers an increased risk of diabetes on her grandchildren. This result was only visible in model 2 (HR = 3.38; CI: 1.18–9.65). A further analysis, using an event of either hospitalization or death from diabetes (underlying or contributory diagnosis) as outcome, failed to confirm this result (Supplementary Table 6). This association should therefore be interpreted with caution.

Supplementary Table 4 suggests that there is no link between parents' access to food and their children's all-cause or cardiovascular mortality. We did observe, however, an elevated diabetes mortality risk among sons in both models (model 2: HR = 1.84; CI: 1.21–2.79), but not among daughters (interaction by gender $p = 0.08$) whose fathers enjoyed good access to food.

**Cancer mortality**. Finally, we examined cancer, a group of causes-of-death not explored in the Överkalix studies, but highly relevant given that it is a leading cause of death in Sweden. Table 3 suggests that if a paternal grandfather had good access to food his male, but not female, grandchildren had a higher risk of dying from cancer (model 2: HR = 3.44; CI: 1.87–6.34). This excess mortality was found both in tobacco-related cancers and cancers not related to tobacco. A highly significant interaction with G2 gender ($p = 0.005$) was observed. (Supplementary Table 3).

In additional analyses, we found no association between paternal grandfather's good access to food and all other (non-cancer) causes-of-death grouped together. Thus, cancer deaths appear to drive the all-cause mortality result for paternal grandfathers in our study.

**Discussion**

We did reproduce one of the main Överkalix results concerning all-cause mortality. For paternal grandfathers with good access to food during their SGP, we find an elevated mortality in grandsons but not in granddaughters. This excess risk among male grandchildren is also highlighted in Pembrey et al.[11].

However, we were unable to reproduce any of their findings for cardiovascular diseases and diabetes. We note, however, that both our replication and the original Överkalix study[10] failed to find an association between grandparental access to food and a grandchild's mortality from CVD.

We could not reproduce their often-quoted finding that paternal grandfathers' exposure to an abundant harvest predicts elevated diabetes mortality in their grandchildren. However, based on 289 deaths, we did find that male offspring of fathers with good access to food were more prone to diabetes. This male-line (father → son) pathway was not observed in the much smaller Överkalix study. Our finding that maternal grandmothers' exposure to food abundance was linked to grandchild diabetes mortality found no support in the Överkalix data.

A key finding of our study concerns cancer, a common cause-of-death in the grandchild generation. We found a clear association between paternal grandfathers with good access to food and their grandsons' mortality from cancer (as a main or contributory cause-of-death). Further, based on 236 cancer deaths, we found a strong interaction between paternal grandfathers' exposure and G2 gender, with men being affected but not women (test for interaction $p = 0.005$).

Following Frankel et al.[27] we separated cancers into tobacco-related and other cancers. Both categories showed an elevated mortality among grandsons of paternal grandfathers with good access to food; among granddaughters cancer mortality was not elevated at all. This gender interaction was driven by cancers not related to tobacco. The use of an alternative classification of cancers[28] gave the same results and served as a sensitivity test.

Some differences between the two studies should be considered. In the Överkalix studies, the mortality effect from parental or grandparental exposure to food shortage or food abundance was always calculated whilst controlling for the exposure of other ancestors. Mostly, only the results after full control of all ancestors were reported. We were only able to take three ancestors into account (all in the paternal or maternal line). A test (see "Method" section) suggests that this difference does not introduce confounding.

Controlling for family social circumstances was done in both studies as a way to rule out the possibility that the observed associations were driven by social or cultural factors. In model 1 (in all tables), we have reproduced the Överkalix studies as closely as possible and followed them in controlling for a number of social circumstances which affected G2 from early in life (see Method section). However, our model 2 differs from the Överkalix method in also controlling for G0 birth year, which is linked both to the likelihood of experiencing a good harvest (later G0 births were more likely to experience good harvests) and to secular trends which influence long-term mortality, such as smoking.

It is also important to take secular trends in the use of fertilizers and pesticides into consideration. During the 1870-1910 period, such practices were still on a low level[29]. When one looks at all significant estimates, they are without exception stronger in models 2 (controlling for G0 birth year) than in models 1. This suggests that our results are not due to confounding from secular trends in smoking or farming practices.

Our G2 study population is 38 times larger than the combined Överkalix cohorts (number of persons at risk) and it is younger, born later. The number of deaths in our G2 is twice as large as in the Överkalix cohorts; the statistical uncertainty in studying grandparental effects (G0 → G2) is therefore smaller. Our G1 population is also considerably larger than that of the Överkalix studies. Our estimates of effects from parent to child (G0 → G1) are based on more than 20 times as many deaths; random errors should therefore be considerably smaller.

The above analysis comprises multiple comparisons. That this may introduce random results is a legitimate concern. We calculated the probability of reproducing, by chance, at least one of the Överkalix results for grandchild mortality, assuming that there are no real associations in either study. There are four

grandparents, each exposed to two events (poor or good harvest), with one outcome (all-cause mortality) in the two genders in G2 plus two outcomes (diabetes and CVD mortality) in the two genders combined. In other words, we replicate 32 analyses in G2. The likelihood that both the Överkalix study and our replication of their analyses will produce a significant result in a particular analysis is $0.05 \times 0.05 = 0.0025$. The chance that both will then point in the same direction (significantly high or significantly low) is 0.5. Thus the likelihood of reproducing a particular result in any one of the 32 G2 analyses should be $32 \times 0.0025 \times 0.5 = 0.04$. In the 24 G1 analyses it is 0.03.

Since the one result that we did reproduce, involving paternal grandfathers, is the one with the strongest a priori backing[11,19], our parallel findings are unlikely to be due to chance. The discrepancies between our two studies concern diabetes and CVD and could be due to randomness or to differences in real-life contexts of the two studies, such as the recording of cause-of-death. Since our population of grandchildren was born in a more recent era, some historical effects may simply not be revealed today. Early death due to infections became more unusual throughout the 20th century; cardiovascular disease in Sweden has been declining since around 1980, while cancer is slowly becoming a more common cause-of-death.

The conclusion in Pembrey et al.[11] and Kaati et al.[12] that a male-line transgenerational effect exists was chiefly based on associations between the paternal grandfather's exposure in SGP and all-cause mortality in his grandsons, and the paternal grandmother's exposure in SGP and all-cause mortality in her granddaughters. In contrast, we found no impact of paternal grandmother's food access on her female offspring's all-cause mortality. The influence of paternal grandfather on all-cause mortality of grandsons, but not granddaughters, was the one key result of the Överkalix studies that we could reproduce.

CVD played no role in this excess mortality among G2 men in either study. Neither did diabetes in our study. We suggest that the excess mortality among male grandchildren whose paternal grandfathers enjoyed good access to food, found in both studies, is at least partly based on cancer mortality. Alternatively, the mortality excess among grandsons could also be of a more general kind, reflecting general susceptibility transmitted across generations.

The finding of a transgenerational response to abundant childhood nutrition in both studies does not prove that the pathway is epigenetic. We refer broadly to epigenetics as heritable changes of gene function not induced by changes in the DNA sequence. Such changes can nevertheless be influenced by DNA sequence. Epigenetic events (such as methylation of DNA) could be genotype-dependent, as shown by several authors, for instance[30,31]. When this is the case we will only be able to observe an average effect, across genotypes.

De novo mutations in the grandparental generation could happen as a response to specific exposures, such as fertilizers, pesticides and mold. Regional differences in harvests could in principle be influenced by differences in the use of fertilizers and pesticides. However, this practice hardly seems common enough at this time[29] to cause new mutations on a large scale. It might have been more difficult to store food during rich harvests when all available storing capacity was exploited. Aflatoxin, a mold thriving in poorly stored food, is a potent carcinogen/mutagen, known to cause mutations in the tumor suppressing *TP53* gene[32]. There is a lack of data concerning these potential exposures in the 1873-1910 period and we can only speculate about their role.

Previous studies of intergenerational effects on offspring health or survival have often focused on starvation or severe food shortage[33,34]. This focus may have been methodologically rewarding but theoretically narrow. Nutritional signals may be

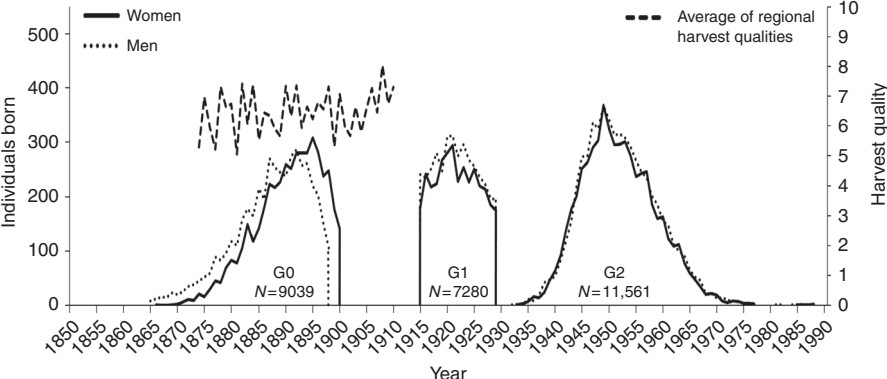

**Fig. 1** Three linked generations. Birth year distribution of G0, G1 and G2 by gender (number of individuals born each year; left *Y*-axis) and annual average harvest quality for years 1874–1910 (right *Y*-axis), which period corresponds to G0 slow growth periods

based on subtle differences in amount and kind of nutrition. In our study, as in the Överkalix studies, it appears that food abundance during the slow growth period is important. Abundant harvests should mean good access to grains and many vegetables, rich in folates and working as methyl donors.

Waterland and Jirtle[35] discuss the importance of dietary methyl donors for DNA methylation. They suggest that "early methyl donor malnutrition (i.e. over- or undernutrition) could effectively lead to premature epigenetic aging, thereby contribute to an enhanced susceptibility to chronic disease in later life" (p.63). The reference to epigenetic aging suggests a vulnerability to disease in general. Two elements in the genome may be especially sensitive to nutritional dysregulation: transposons and imprinted genes[30]. Imprinted genes are usually marked for parent-of-origin in sperm or ova, in sperm thus marked by father's experience.

Epigenetics play a central role in neoplasia[36–38]. Numerous small ncRNAs are exclusively or preferentially expressed in testis or germ cells in humans and mice[39]. Reddy[40] concluded that aberrant miRNA expression is a rule rather than an exception in carcinogenesis. Hypermethylation of CpG islands upstream from tumor suppressor genes would influence cancer risk[41]. The silencing of tumor suppressor miRNAs contributes to the development of human cancer and metastasis[42]. A transgenerational response to abundant childhood food could thus be based either on epigenetic mechanisms linked to nutrition, or de novo mutations linked to new farming practices, or both together since these mechanisms are not mutually exclusive.

The so-called Carnegie survey of family diet and health in prewar Britain collected very detailed information about children's food intake in 1937-39. Frankel et al.[27]. followed the children up to 1996 with regard to mortality. Controlling for social factors they observed that boys and girls consuming rich food (highest fifth of energy content) were more than twice as likely to die of cancer later in life compared to the lowest fifth. This effect was particularly strong for cancer not related to tobacco. In their view, it was the high energy content of the food that gave rise to this effect rather than any mutagenic or carcinogenic substances in food. Still, assuming that the number of cells in (organs of) the body respond to abundant nutrition early in life, the risk of de novo mutations due to chance would increase in proportion to cell numbers. Consistent with this hypothesis, obesity[43], body height[44] and number of (stem) cells in specific organs[45,46] have all been linked to cancer mortality.

Abundance of food, when the body is just about to leap into puberty, i.e., an energy demanding development, may also influence genome-wide DNA methylation patterns in boys and

girls, including in germ-line DNA, and impact on small non-coding RNAs in sperm cells.

Soubry et al.[13] suggested four periods of susceptibility, when a change of epigenetic patterns in male germ cells would be possible. One of them is the period just before puberty, equivalent to the slow growth period, which is the period we have examined in this replication study.

Could nutritional experience in one generation thus trigger a transgenerational response in subsequent generations? Our results lend support for the existence of a male-line transgenerational pathway, triggered by events during the paternal grandfather's slow growth period. We would like to be cautious about the specific mechanism. However, the hypothesis that a molecular signal of abundant nutrition received in this period could be captured by male gametes, cannot be rejected and should thus be further explored[8]. The implications for our understanding of ourselves are substantial[47].

## Methods
**Study population.** The Uppsala Birth Cohort Multigeneration Study (UBCoS Multigen) started with manually tracing, in archives, all births at Akademiska Hospital in Uppsala 1915–1929 ($n = 14,612$). Its multigenerational extension includes the 12,168 individuals in the first generation (G1) who have later been identified by their personal ID number. We traced their (now deceased) parents (G0) manually and their children (G2) through linking to the Swedish Multigenerational Registry. The other parent of a G2 child was not included in the original UBCoS Generation 1. This means that we usually have information about paternal or maternal grandparents, but not about all four. Figure 1 illustrates the three linked generations, and the number of members in each generation in this particular study. The multigenerational data base is suitable for our purpose[48,49]. The statistical power to detect effects of the same size as those reported from the Överkalix cohorts is reported in Supplementary Table 7.

Information about G0 was obtained from the original birth records of G1 and from information collected from parish and hospital registers. Their places of birth were collected from the sixth edition of the Swedish Death Index, published by Statistics Sweden and the Swedish Genealogical Society. G0 consists of 15,706 individuals, born 1851–1914, the overwhelming majority born outside the city of Uppsala. In this study, we include those born from 1865 to 1900 ($n = 9039$) for whom we have collected harvest data. Our main study focus is the grandchildren (G2) of these 9039 individuals. G2 includes all of G1's children born 1932 or later who had not died by the start of 1961. Adopted children ($n = 381$), individuals whose personal identity number has been reused and those whose death date was obviously wrong ($n = 48$) were excluded. Since we restricted the analyses to G2 individuals for whom we had harvest information for both maternal or both paternal grandparents, we were able to analyze outcomes in 11,561 grandchildren.

In additional analyses, we studied G1 mortality in 3820 men and 3460 women by their G0 mothers' and fathers' food access.

Linking of individuals to patient and mortality data was made by Statistics Sweden and all analyses were made on data anonymized to researchers. This follows standard practice in large registry-based observational studies in Sweden. The Regional Ethical Review Board of Stockholm approved of all aspects of the study (dnr 2015/904-3115; dnr 2016/933-32).

**Exposure data: contextual variation approach**. Harvest variations between years and regions across Sweden were substantial. Exposure in G0 (access to food) was defined from this variation. This is referred to as an "intention to treat instrumental variable approach" in the econometrics literature[50].

There were 24 regions in all, based on the administrative unit (län). In the year 1890, the population of Sweden was 4.8 million, so an average region covered around 200,000 inhabitants. Most grandparental sample members (around 55%) are from the region around Uppsala (Uppsala Län), which counts among the smaller regions and contained about 120,000 inhabitants. There would be a certain variation in harvests within all regions. Trade within regions would have reduced the importance of these differences, as administrative divisions often followed trade and commercial patterns. This would certainly have reduced the within-region differences in food access.

The regional average in a certain year would in all likelihood be an underestimation of actual food consumption in rich families and an overestimation in poor families. We controlled for social factors as far as possible in all analyses, which would have reduced any social bias.

Finally, excluding city dwellers gave stronger estimates of effect, suggesting that actual food consumption was better captured among rural inhabitants.

Our measure of access to food was based on an annually published regional harvest statistic for 1874–1910[51] ranging from 0 (total crop failure) to 10 (abundant harvest). Each G0 individual was assigned an access to food value, year by year during his/her slow growth period (SGP, defined as ages 9–12 for boys and 8–10 for girls), based on that statistic for his/her region of residence.

A weighting procedure was applied, based on the assumption that harvests took place on September 1st. G0 entering their SGP in September were considered to rely fully on the current year's harvest. Those with birthdays later in the year relied partly on the subsequent year's harvest (current year's harvest × 11/12 + next year's harvest × 1/12 for those born in October). For those with birthdays earlier in the year a corresponding dependence on the previous year's harvest was assumed. A weighted score of less than 5 ('a fourth below typical harvest') was considered a poor harvest and a weighted score of 8.5 ('more than good, unusually good') or more was considered a good harvest. Moderate harvests were thus defined as 5–8.4; poor as 0–4.9 and good as 8.5–10.

G0 were then classified into one of three mutually exclusive exposure groups, who (1) experienced at least one good harvest and no poor harvest during SGP ($n = 478$) (2) experienced only moderate harvests during SGP (reference category, $n = 8,142$), or (3) experienced at least one poor harvest and no good harvest during SGP ($n = 419$). No G2 individual in the study population had a grandparent who experienced both good and poor harvests in SGP.

To control for parental (G1) food access in analyses of the effects of grandparental (G0) food access on G2 mortality we had to rely on annually published national harvest statistics, thereby losing precision. A more recent period (1923–1941) also meant less dramatic variation in exposure. This statistic ranged from 1 (poor) to 5 (good), with 3 being average harvest. We applied the same weighting procedure as for G0, and generated a binary variable for experiencing good harvest in SGP (a weighted score of more than 3.3 any year, $n = 2,631$) or not (a weighted score of 3.3 or less for all years, $n = 2,713$).

Region of residence at G0 birth was known for all included G0, and used as a proxy for residence during SGP. A proportion of G0 moved between birth and age ten. In a random sample of G0 ($N = 211$) we tested and confirmed that our results concerning grandparental food abundance are not due to selective out-migration.

**Outcome data: mortality and hospitalization**. G2 mortality data for 1961–2015 was obtained from the Swedish Cause-of-Death Register. In total, 1255 (10%) of G2 died during follow-up. CVD, diabetes or cancer as underlying or contributing cause of death were used when analyzing cause-specific mortality. CVD corresponds to ICD-9 diagnoses 390–459; diabetes to code 250 and cancer to codes 140–209. In additional analyses of G2 outcomes, we also used data from the Swedish In-patient Register (1961–2015). An event was defined as a first case of hospitalization, or a death. Since several disease entities may be entered on a death certificate or on a hospital discharge note, individuals may contribute events to more than one cause-specific analysis. Our analyses of all-cause mortality include two individuals with known death date but unknown cause-of-death.

**Control for confounding**. In all regression analyses we controlled, as closely as possible, for the same social and demographic factors as in the Överkalix studies. Models 1 in Tables 1, 2, Supplementary Table 1, Supplementary Table 2 and Supplementary Table 4 could therefore be regarded as a replication of these studies. In model 2 we include further controls. Table 3, Supplementary Table 3, Supplementary Table 5 and Supplementary Table 6 and models 2 everywhere represent further explorations.

We considered food access of other ancestors. Our grandchild mortality analysis could take into account the food access of three ancestors simultaneously—all three on either the paternal or the maternal side. A small number of G1 members had children together; for those G2 individuals ($N = 605$) we could compare estimates adjusted for three and six ancestors. There were virtually no differences between the two. Thus, estimates of the effect on G2 mortality from, say, paternal grandfather's or maternal grandmother's food access, presented in tables, are unlikely to be confounded by the effect of any other ancestor's food access.

Social and demographic confounders were considered. G2 year of birth was grouped into five-year age bands as a categorical variable in all tables, in model 1 and 2. Model 1 controlled for a number of social variables. These were mother's (maternal lineage) or father's (paternal lineage) highest achieved education, collected from the Swedish Census 1970, grouped into elementary or more than elementary education; family income, obtained from the same source, grouped into quintiles based on a couple's total earned income; social class, from the Swedish Census 1960 (non-manual workers, manual workers, farmers and entrepreneurs, and unknown); mother's parity, which defined sibling position of G2 as 1, 2, 3–4, 5–6 or 7 and higher; sibship size; and finally, whether a parent died before the child was 18.

G0 birth year and common ancestors were also considered. Thus, models 2 controlled for G0 birth years (as linear trends) and cluster standard errors at family level, to account for the fact that siblings and cousins share biological ancestors. In analyses of G1 mortality we controlled for G1 birth year (5-year groups), G1 family social class (six groups) and marital status at birth plus sibling position (defined as in G2 analyses).

**Models**. In all G1 and G2 mortality analyses, we compared G0 individuals who differed with respect to SGP ancestral food access, controlling for the above confounders in regression analyses. Hazard ratios (HR) were estimated by Cox proportional hazard models with age as underlying time scale. For all HRs 95% confidence limits are reported; analyses of interaction between food access and gender were performed by introducing a product term and calculating a corresponding p-value with Stata 14.2. All models accounted for censoring. Left censoring arose, because our G2 data do not include individuals who died before 1 January 1961. For G1 mortality follow-up starts at 1 January 1952 (earliest data with digitalized cause-of-death data). Right censoring arose if individuals emigrated or survived beyond 31 December 2015. All analyses were performed using Stata 14.2.

## Data availability

The data that support these findings are available on reasonable request to the corresponding author [DV]. The availability of social, patient and mortality data is subject to restrictions imposed by the National Board of Health and Welfare and Statistics Sweden, in accordance with Swedish legislation on privacy protection, meaning that data can only be accessed and analyzed at a special venue in Stockholm. A Reporting Summary is available as a supplementary information file.

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

## Acknowledgements

Bitte Modin for initial discussions and management; Samantha Kearney for excellent background reading; Corinna Hartung for exploratory analyses of incomplete data; Agneta Cederström for data management; Dave Leon and Mats Lambe for discussions on cancer; Jenny Kreuger for discussions on aflatoxin and the use of chemicals in Swedish farming; Marcus Pembrey for helpful discussions about epigenetics. The Swedish Research Council (K2015-69 × -22720-01-3) funded the project.

## Author contributions

D.V. and G.J.v.B. designed the study. D.V., P.P., V.A. and G.J.v.B. performed research. P.P. and V.A. analyzed data. D.V., P.P., V.A. and G.J.v.B. wrote the paper.

## Additional information

**Competing interests:** The authors declare no competing interests.

