## [Peer Review File · Nature Communications]

Reviewer #1 (Remarks to the Author):

This is a well-written manuscript. A pleasure to read. In the analysis of data for the Uppsala Multigeneration Study, the authors used harvest data during the pre-pubertal (the slow growth) period of grandparents to evaluate associations with all-cause and disease specific mortality (diabetes, cardiovascular disease risk and cancer) in children. They report associations between higher food access and all-cause mortality in male grandchildren and associations appear to be driven by cancer, that may not be tobacco related. Exposure assessment was constructed from regional harvest statistics and outcomes were based on population-based disease registers.

The data are important as there are few human data demonstrating potential effects of excessive food access (common in most societies today) and transgenerational effects of common chronic diseases, including the top two causes of mortality globally, cardiovascular diseases and cancer. The sample size is large enough to make a reasonable case for excessive caloric input increase the risk of multiple cancers.

Some weaknesses that the authors will need to address to improve the interpretation of these intriguing findings.

1. Historical harvest data are used to construct the 'access to food exposure' variable. For transparency, how large these regions and the extent to which averages represent the actual exposure of individuals need to be discussed.
2. While the authors have been thoughtful in thinking through potential cohort effects, not discussed is the potential effects of changes in diet and attendant chemicals in food given rapid changes in agricultural practices (e.g., pesticides, fertilizers that contained large amounts of carcinogens over the pre-pubertal slow growth subsequent generations). An analysis that directly addresses this or a discussion of why this is unlikely to change their findings is warranted.
3. The analysis separating cancers by 'smoking vs. not-smoking related' should be reconsidered as half of the cancers in the Frankel paper have smoking as an established risk factor, e.g., cervical, prostate, GI cancer (e.g., liver, esophageal-upper and lower-, and pancreas) cancer.

Specific

In the abstract, there is a need for sample size.

Reviewer #2 (Remarks to the Author):

This is a very interesting paper that tests, in a dataset 38 times larger, the hypotheses of the often-cited Overkalix study, with additional new analyses looking at cancer. It will be very important in the field of developmental programming, and potentially epigenetics. The paper is very well written and exceptionally concise, which I see as a strength (the paper is not lacking in detail or analyses). I recommend that it is published after consideration of the points below:

In the Introduction, I suggest changing "Their findings, have been discussed in the best journals..." to remove the comma after "findings" and consider changing "best" to something like "most famous".

In the Results, it might be useful to conduct a power calculation, showing that, if an association with the same effect size as identified in Overkalix DID exist, this study had XX% power to detect it (or alternatively, the smallest estimated effect that this study has 80% power to detect, and how that relates to the effect estimates reported in the Overkalix study).

In the Discussion, "This is consistent with a male-line (father  son) epigenetic pathway", since there is no epigenetic data in either the Overkalix study or this study, I think this statement is too strong.

As a reader unfamiliar with the study by Frankel et al, a brief description of why this study is cited as the reason for separating tobacco-related and other cancers would be useful. This is explained towards the end of the Discussion, but a brief mention further up would be helpful.

"Associations visible after control for family social circumstances are more likely to be epigenetic" - I think caution is needed with regards to the definition of "epigenetic" used in this paper. It appears that the term is being used broadly to mean a biological effect caused by an environmental exposure to a previous generation, whereas there are in fact multiple more complex definitions. This is one of the criticisms of the original Overkalix papers, and indeed papers on developmental programming more widely. Without epigenetic data, these claims are not supported. Could the associations found here represent confounding by genetic factors (including de novo genetic mutations)? Or other elements of gene regulation that aren't considered "epigenetic"? This is the main oversight/area of weakness in this otherwise excellent paper, but until addressed, I see it as substantial.

"Our results give support for the existence of a male-line transgenerational pathway, which is open during the slow growth period" - do the authors have any insights or opinions on whether this pathway is open during other periods? Also, if it exists, why do the effects appear to skip a generation?

Responses to reviewers.

Reviewer #1

This is a well-written manuscript. A pleasure to read. In the analysis of data for the Uppsala Multigeneration Study, the authors used harvest data during the pre-pubertal (the slow growth) period of grandparents to evaluate associations with all-cause and disease specific mortality (diabetes, cardiovascular disease risk and cancer) in children. They report associations between higher food access and all-cause mortality in male grandchildren and associations appear to be driven by cancer, that may not be tobacco related. Exposure assessment was constructed from regional harvest statistics and outcomes were based on population-based disease registers.

The data are important as there are few human data demonstrating potential effects of excessive food access (common in most societies today) and transgenerational effects of common chronic diseases, including the top two causes of mortality globally, cardiovascular diseases and cancer. The sample size is large enough to make a reasonable case for excessive caloric input increase the risk of multiple cancers.

-We are very grateful for this positive assessment of our paper and for the helpful comments, which we respond to below.

Some weaknesses that the authors will need to address to improve the interpretation of these intriguing findings.

1. Historical harvest data are used to construct the ‘access to food

exposure' variable. For transparency, how large these regions and the extent to which averages represent the actual exposure of individuals need to be discussed.

-We have added the following text in the Methods section:

“There were 24 regions in all, based on the administrative unit (län). In the year 1890 the population of Sweden was 4.8 million, so an average region covered around 200,000 inhabitants. Most grandparental sample members (around 55%) are from the region around Uppsala (Uppsala Län), which counts among the smaller regions and contained about 120,000 inhabitants. There would be a certain variation in harvests within all regions. Trade within regions would have reduced the importance of these differences, as administrative divisions often followed trade and commercial patterns. This would certainly have reduced the within-region differences in food access.

The regional average in a certain year would in all likelihood be an underestimation of actual food consumption in rich families and an overestimation in poor families. We controlled for social factors as far as possible in all analyses, which would have reduced any social bias.

Finally, excluding city dwellers gave stronger estimates of effect, suggesting that actual food consumption was better captured among rural inhabitants”.

2. While the authors have been thoughtful in thinking through potential cohort effects, not discussed is the potential effects of changes in diet and attendant chemicals in food given rapid changes in agricultural practices (e.g., pesticides, fertilizers that contained large amounts of carcinogens over the pre-pubertal slow growth subsequent generations). An analysis that directly addresses

this or discussion of why this is unlikely to change their findings is warranted.

-Thank you for raising this point. The slow growth period of the grandparental generation corresponds to the period 1873-1910. In this period the usage of chemical fertilizers and pesticides was still low. After 1910, Swedish farming practices changed to a more frequent use of fertilizers and pesticides. This would affect exposure of the second and third generation, but would not be related to our exposure variable for the first generation (G0).

We now mention this in the text, for instance:

“It is also important to take secular trends in the use of fertilizers and pesticides into consideration. During the 1870-1910 period, such practices were still on a low level [29]. When one looks at all significant estimates, they are without exception stronger in models 2 (controlling for G0 birth year) than in models 1. This suggests that our results are not due to confounding from secular trends in smoking or farming practices”.

(Discussion, page 7).

And further (Discussion, page 10):

“De novo mutations in the grandparental generation could happen as a response to specific exposures, such as fertilizers, pesticides and mold. Regional differences in harvests could in principle be influenced by differences in the use of fertilizers and pesticides. However, this practice hardly seems common enough at this time [29] to cause new mutations on a large scale. It might have been more difficult to store food during rich harvests when all available storing capacity was exploited. Aflatoxin, a mold thriving in poorly stored food, is a potent carcinogen/mutagen, known to cause mutations in the tumor suppressing TP53 gene [32]. There is a lack of data concerning these potential exposures in the 1873-1910 period and we can only speculate about their role”.

3. The analysis separating cancers by ‘smoking vs. not-smoking related’ should be reconsidered as half of the cancers in the Frankel paper have smoking as an established risk factor, e.g., cervical, prostate, GI cancer (e.g., liver, esophageal-upper and lower-, and pancreas) cancer.

-We have reconsidered this issue. Classifying cancers into tobacco-related and not tobacco-related is problematic in many ways. The degree to which specific types of cancer are tobacco-related varies enormously. And tobacco-related cancers may at the same time be related to other risk factors. Liver and stomach cancers, for instance, may be related to both food/nutrition and tobacco. Frankel, in fact, classifies both as not tobacco-related. In contrast, a later classification, used by Wirdefeldt et al (ref [28]: Am J Epidemiol 2013) classifies both liver and stomach cancer as tobacco-related. We made a sensitivity analysis, using Wirdefeldts scheme instead of that of Frankel. Although we found that HR estimates differ somewhat between classifications, the main result remains: cancer mortality not related to tobacco is elevated among the grandsons, but not the granddaughters, of paternal grandfathers with good access to food. The inevitable conclusion is that tobacco consumption is not driving our results on cancer. For consistency, we prefer to use Frankel’s classification when comparing his results, within one generation, with our results across generations.

We now mention this sensitivity test in the text and gives the reference to Wirdefeldt et al.

4. Specific

In the abstract, there is a need for sample size.

-There are 9,039 individuals in the grandparental (G0) generation,

7,280 in G1 and 11,561 in G2 (effective samples). We have now added these figures to the abstract.

Reviewer #2

This is a very interesting paper that tests, in a dataset 38 times larger, the hypotheses of the often-cited Overkalix study, with additional new analyses looking at cancer. It will be very important in the field of developmental programming, and potentially epigenetics. The paper is very well written and exceptionally concise, which I see as a strength (the paper is not lacking in detail or analyses). I recommend that it is published after consideration of the points below:

-Thank you for this positive assessment and for the constructive comments below. We believe that the manuscript is now further improved.

In the Introduction, I suggest changing "Their findings, have been discussed in the best journals..." to remove the comma after "findings" and consider changing "best" to something like "most famous".

-We have changed the text in line with this suggestion.

In the Results, it might be useful to conduct a power calculation, showing that, if an association with the same effect size as identified in Overkalix DID exist, this study had XX% power to detect it (or alternatively, the smallest estimated effect that this

study has 80% power to detect, and how that relates to the effect estimates reported in the Överkalix study).

-We have now performed power calculations for overall mortality and for diabetes and cardiovascular mortality respectively, in 10 instances where the Överkalix studies report significant ($p < 0.05$) or near-significant ($p < 0.10$) results. The results provide an answer to the question above: if an association of the same effect size as that identified in Överkalix did exist, what would be the power in our study to detect it? First, analysing G2 mortality by grandparental (G0) exposure, we found a statistical power of between 72% and 99% to detect such effects. Second, in analyzing G1 mortality by parental (G0) exposure, we had a power of over 99% in all three instances where Överkalix studies showed a significant, or close to significant, effect. This is now clear from the second paragraph in the Results section and from Supplementary Table S7.

In the Discussion, "This is consistent with a male-line (father  son) epigenetic pathway", since there is no epigenetic data in either the Överkalix study or this study, I think this statement is too strong.

-We have now rephrased this sentence as follows: "This male-line (father→son) pathway was not observed in the much smaller Överkalix study".

As a reader unfamiliar with the study by Frankel et al, a brief description of why this study is cited as the reason for separating tobacco-related and other cancers would be useful. This is explained towards the end of the Discussion, but a brief mention further up would be helpful.

-There is now a brief mention in the Introduction.

It was the conclusion by Frankel et al., that the energy content in food in childhood, carefully measured as the highest quintile of energy intake, may influence cancer risk, that inspired us to ask whether this could also trigger a transgenerational effect (in boys). Their separation into tobacco-related and other cancers allowed them to rule out confounding from smoking, i.e the possibility that their finding was due to higher levels of smoking among people with high food consumption early in life.

See also comment 3 to Reviewer 1 above.

"Associations visible after control for family social circumstances are more likely to be epigenetic" - I think caution is needed with regards to the definition of "epigenetic" used in this paper. It appears that the term is being used broadly to mean a biological effect caused by an environmental exposure to a previous generation, whereas there are in fact multiple more complex definitions. This is one of the criticisms of the original Overkalix papers, and indeed papers on developmental programming more widely. Without epigenetic data, these claims are not supported. Could the associations found here represent confounding by genetic factors (including de novo genetic mutations)? Or other elements of gene regulation that aren't considered "epigenetic"? This is the main oversight/area of weakness in this otherwise excellent paper, but until addressed, I see it as substantial.

-We are grateful for this comment. We have changed the quoted sentence. As you pointed out this is an issue that goes beyond our

paper in that it concerns how we talk about the relation between cultural, epigenetic and genetic systems.

We assume that G0 food access, measured by a regional harvest indicator, is independent of any genetic characteristics in G0. However, genetic factors certainly take part in the regulation of any metabolic response to G0 abundant nutrition. We realise that this could be important for heterogeneity in cancer susceptibility in the grandchild generation. In such case, we would only be able to observe an average effect across genotypes.

We have added the following to the Discussion section (page 9):

“The finding of a transgenerational response to abundant childhood nutrition in both studies does not prove that the pathway is epigenetic. We refer broadly to epigenetics as heritable changes of gene function not induced by changes in the DNA sequence. Such changes can nevertheless be influenced by DNA sequence. Epigenetic events (such as methylation of DNA) could be genotype-dependent, as shown by several authors, for instance [30; Hannum et al] [31; Lan et al]. When this is the case we will only be able to observe an average effect, across genotypes”.

About de novo mutations we have added this: (Discussion, page 10):

“De novo mutations in the grandparental generation could happen as a response to specific exposures, such as fertilizers, pesticides and mold. Regional differences in harvests could in principle be influenced by differences in the use of fertilizers and pesticides. However, this practice hardly seems common enough at this time [29] to cause new mutations on a large scale. It might have been more difficult to store food during rich harvests when all available storing capacity was exploited. Aflatoxin, a mold thriving in poorly stored food, is a potent carcinogen/mutagen, known to cause

mutations in the tumor suppressing TP53 gene [32]. There is a lack of data concerning these potential exposures in the 1873-1910 period and we can only speculate about their role”.

We also considered de novo mutations due to chance. Frankel’s paper showed a within-generation effect on cancer of abundant childhood food. We write that

“In their view, it was the high energy content of the food that gave rise to this effect rather than any mutagenic/carcinogenic substances in food. Still, assuming that the number of cells in (organs of) the body respond to abundant nutrition early in life, the risk of de novo mutations due to chance would increase in proportion to cell numbers”. (page 11).

We acknowledge that : “A transgenerational response to abundant childhood food could thus be based either on epigenetic mechanisms linked to nutrition, or on de novo mutations linked to new farming practices, or on both together since these mechanisms are not mutually exclusive” (page 11).

And conclude that:

“Our results give support for the existence of a male-line transgenerational pathway, triggered by events during the paternal grandfather’s slow growth period. We would like to be cautious about the specific mechanism. However, the hypothesis that a molecular signal of abundant nutrition received in this period could be captured by male gametes, cannot be rejected and should thus be further explored” (page 11)

See also response 2 to Reviewer 1.

"Our results give support for the existence of a male-line transgenerational pathway, which is open during the slow growth

period" - do the authors have any insights or opinions on whether this pathway is open during other periods?

-We have no special insights but we are aware of the work of Soubry (ref 13) and Wu (ref 14). Apart from the immediate period after fertilization when epigenetic reprogramming takes place, both suggest that each reproductive cycle, from puberty onwards, could be relevant. In this paper, we have only tested whether the pre-pubertal period is one of susceptibility. This follows from our aim to replicate the Överkalix studies. The authors of the Överkalix studies argue that the slow growth period of a child may be the sensitive period for epigenetic modifications of male gametes. The SGP is associated with the emergence of the first viable pools of spermatocytes and the beginning of re-programming of methylation imprints (Pembrey EJHG 2002).

This period is also found to be critical for the development of health related characteristics, such as height, in within-generation studies. Van den Berg et al (JEAA 2014) has an extensive discussion.

Also, if it exists, why do the effects appear to skip a generation??

-The phenomenon was observed in our study as well as in the Överkalix study. It is discussed very briefly by those authors. Bygren, in their first paper (2001), considers this phenomenon to be “consistent with genomic imprinting” and to “constitute a feedforward control loop”. We can only speculate at this point. We would expect a new mutation in G0 to be expressed in both G1 and G2 and perhaps in both genders. But an “epi-mutation” in a G0 male germ cell could perhaps be offset in G1 soma by other signals of opposite effect, and still be carried forward to G2.

We have not speculated about this observation in this manuscript.

Reviewer #1 (Remarks to the Author):

Thank you for the full response to my concerns and the revision of the manuscript. I think this paper will make an important contribution as we better understand the role of the paternal germline in the early development of cancer.

I recommend its publication with no further delay

Reviewer #2 (Remarks to the Author):

The authors have done a very comprehensive job of addressing my concerns. I am very happy to recommend acceptance of this paper.

Response to Reviewers comments/ Vågerö et al NCOMMS 18-15212A

REVIEWERS' COMMENTS:

Reviewer #1 (Remarks to the Author):

Thank you for the full response to my concerns and the revision of the manuscript. I think this paper will make an important contribution as we better understand the role of the paternal germline in the early development of cancer.

I recommend its publication with no further delay

-Thank you

Reviewer #2 (Remarks to the Author):

The authors have done a very comprehensive job of addressing my concerns. I am very happy to recommend acceptance of this paper.

-Thank you